# Treatment of corneal endothelial damage in a rabbit model with a bioengineered graft using human decellularized corneal lamina and cultured human corneal endothelium

Francisco Arnalich-Montiel[1], Adrian Moratilla[2], Sherezade Fuentes-Julián[2], Veronica Aparicio[2], Marta Cadenas Martin[2], Gary Peh[3], Jodhbir S. Mehta[3], Khadijah Adnan[3], Laura Porrua[1], Ane Pérez-Sarriegui[1], Maria P. De Miguel [2]*

1 Ophthalmology Department, Hospital Ramón y Cajal, Madrid, Spain, 2 Cell Engineering Laboratory, La Paz Hospital Research Institute, iDIPAZ, Madrid, Spain, 3 Singapore Eye Research Institute, Singapore, Singapore

* mariapdemiguel@gmail.com

## Abstract

### Objective

We aimed to investigate the functionality of human decellularized stromal laminas seeded with cultured human corneal endothelial cells as a tissue engineered endothelial graft (TEEK) construct to perform endothelial keratoplasty in an animal model of corneal endothelial damage.

### Methods

Engineered corneal endothelial grafts were constructed by seeding cultured human corneal endothelial cell (hCEC) suspensions onto decellularized human corneal stromal laminas with various coatings. The functionality and survival of these grafts with cultured hCECs was examined in a rabbit model of corneal endothelial damage after central descemetorhexis. Rabbits received laminas with and without hCECs (TEEK and control group, respectively).

### Results

hCEC seeding over fibronectin-coated laminas provided an optimal and consistent endothelial cell count density and polygonal shape on the decellularized laminas, showing active pump fuction. Surgery was performed uneventfully as standard Descemet stripping automated endothelial keratoplasty (DSAEK). Corneal transparency gradually recovered in the TEEK group, whereas haze and edema persisted for up to 4 weeks in the controls. Histologic examination showed endothelial cells of human origin covering the posterior surface of the graft in the TEEK group.

**Data Availability Statement:** All relevant data are within the manuscript and its Supporting Information files.

**Funding:** This study was supported in part by grants EC-11 and PI15/01447 from the Ministry of Health and Social Politics, Spain; PI15/01447 was supported by the Instituto de Salud Carlos III (Plan Estatal de I+D+i 2013-2016) and cofinanced by the European Development Regional Fund "A way to achieve Europe" (ERDF); and financing from Roche Farma SA; AP2010-0659 fellowship to SF-J from the Ministry of Education, Culture and Sports, Spain; and from the BioMedical Foundations Mutua Madrileña and Marato TV3, Spain. V Aparicio, M Cadenas and A Moratilla were supported by the Consejería de Educación, Juventud y Deporte of Comunidad de Madrid and by Fondo Social Europeo (Programa Operativo de Empleo Juvenil, and Iniciativa de Empleo Juvenil (YEI), (PEJ15/BIO/AI/0045, PEJD-2018-PRE/BMD-8878, PEJ15/BIO/AI/0093 and PEJD-2018-PRE/BMD-9040).

**Competing interests:** The authors haave declared that no competing interests exist.

## Conclusions

Grafting of decellularized stroma carriers re-surfaced with human corneal endothelial cells *ex vivo* can be a readily translatable method to improve visual quality in corneal endothelial diseases.

## Introduction

Corneas are the most commonly transplanted tissue worldwide; in 2012, 184,577 corneal transplants were performed in 116 countries but tissue was only procured in 82 countries [1]. Despite these high numbers, eye banks cannot match the demand worldwide and it is estimated that there is 1 cornea available per 70 needed [1].

It is clearly necessary to encourage corneal donation in all countries [2], but alternative and complementary developments are currently being explored to overcome the shortage of corneas, including artificial or bioengineered corneas, as well as genetic or medical manipulation of human corneal cells to promote proliferation and migration [2,3].

Corneal bioengineering using expanded human corneal endothelial cells (hCECs) appears to be a particularly feasible and viable technique in the short term for supplying extra tissue for endothelial keratoplasty (EK)[4,5]. EK represents over one-third of all corneal grafts performed and involves replacing the damaged or pathological corneal endothelium. The corneal endothelium is a monolayer cell sheet that coats the inner surface of the cornea and regulates corneal hydration and therefore transparency. Unlike epithelial corneal cells, human endothelial cells are quiescent *in vivo*, and had proven extremely difficult to expand *in vitro* until recently [6–11].

The ideal bioengineered endothelial graft would be comprised of a thin, transparent, adherent and corneo-biocompatible carrier, seeded with functioning corneal endothelial cells (CECs) that is capable of being inserted into the eye through a small incision [12]. Current endothelial tissues used for EK, including both Descemet membrane (DM) alone for Descemet membrane endothelial keratoplasty (DMEK) or Descemet membrane along with a thin stromal tissue for Descemet stripping automated endothelial keratoplasty (DSAEK), have shown their effectiveness in more than 10 years of worldwide experience [13]. Endothelial graft bioengineering with stromal lamellae as a carrier mimics the tissue used for DSAEK and is more easy to handle than DM alone and more controlled delivery than cell injection [14], hence has been proposed for expanding the new technique to a large population [12,15–17].

We recently reported the clinical trial use of human decellularized stromal laminas in a corneal pocket as a biocompatible and safe implant to treat advanced keratoconus [18]. Thus, we investigated the feasibility of these human decellularized stromal laminas seeded with human corneal endothelial cells as an endothelial graft construct to perform endothelial keratoplasty in an animal model of corneal endothelial damage.

## Materials and methods

### In vitro preparation of decellularized corneal stroma lamellae

The Ethics Committee of La Paz University Hospital, as well as the Community of Madrid (Spain) Ethics Committee for Animal Research approved the study. Animal studies were performed in compliance with the Association for Research in Vision and Ophthalmology (ARVO) statement for the use of animals in ophthalmic and vision research. All the human

tissue used had consent for use in research, and the Helsinki Declaration for biomedical research involving human subjects was adhered to throughout the study. None of the transplant donors were from a vulnerable population, and all donors or next of kin provided written informed consent that was freely given.

To obtain stromal lamina, 7 corneas from 25 to 40 years old donors were stored in Optisol GS (Bausch & Lomb Incorporated, Rochester, NY, USA) for five to eleven days at 4˚C until DMEK for clinical use, and the remaining corneoscleral button, denuded of DM, was acquired for our study. Corneal tissue was then trephined with an 8-mm diameter corneal Hanna vacuum punch (Moria Surgical, Antony, France) and frozen at -80˚C in optimal cutting temperature (OCT) compound (Tissue-Tek, Miles Laboratories, Naperville, IL, USA). Corneas were flatten manually previous to freezing. Tissue samples were cut at 150 µm on a cryostat throughout their thickness, giving 3 to 6 laminas per donor cornea depending on the thickness. Anterior or posterior laminas were used in a blind fashion. Laminas underwent a decellularization process as previously published, which includes incubation with sodium dodecyl sulfate and DNAse, and washes with phosphate-buffered saline (PBS) +1% antibiotics [19]. Graft transparency was assessed by placing the lamellae on a back-lit chart as in the He group study [12]. Laminas maintained a 90% transparency and the 8 mm diameter after decellularization.

Decellularization was confirmed by cutting frozen OCT compound-embedded sections at 8-µm thickness, placement on silane-coated microscope slides (Muto, Tokyo, Japan), staining with hematoxylin-eosin and observation with light microscopy (Carl Zeiss AG, Dublin, IR) or by 4',6-diamidino-2-phenylindole (DAPI) staining and observation under fluorescence microscopy (Carl Zeiss AG, Dublin, IR).

### Endothelial cell seeding

Corneoscleral buttons of 16-mm diameter trephination, unsuitable for transplantation, were obtained from the eye bank of the Lions Eye Institute, Tampa, FL, USA. 32 corneas from 16 donors were used for the study, and the information of the donors and tissues is provided in Table 1. Corneas were stored in Optisol-GS storage medium until use (Table 1).

Corneal endothelium and DM were peeled off with watchmaker's tweezers (Roboz) and incubated with 2 mg/ml collagenase type I at 37˚C for 4 hours. When cells separated from the DM, they were centrifuged at 1000g, incubated with trypsin-like TripLE solution (5 minutes), centrifuged again in a similar way and then seeded on tissue culture plates following the protocol of Peh et al. [9] until reaching confluence (typically 3–4 weeks). Briefly, hCEC were seeded at $6 \times 10^2/mm^2$ in fibronectin coated plates in M4 medium consisting of 50% Ham's F12 media, 50% M199, 5% fetal bovine serum (FBS), 2mg/ml ascorbic acid, ITS (10 ug/ml insulin; 5,5 ug/ml transferrin and 5 ng/ml selenium), 1% penicillin/streptomycin (P/S) and 250 ng/ml amphotericin B until reaching confluence. Cultures were then maintained in M5 medium consisting of Human Endothelium Serum Free Medium (Gibco), 5% FBS, 1% P/S and 250 ng/ml amphotericin B.

Then, laminas were recellularized by centrifugation at 200g of passage 1 hCECs directly on the tissue culture wells at $3 \times 10^3$ cells/mm$^2$ for different durations, that is, 10 min or 1 h to improve endothelial cell adhesion. In order to ascertain which extracellular matrix coating would further improve endothelial cell adhesion and maintenance of the phenotype, decellularized corneal laminas were coated with either fibronectin (FNC), collagen IV, Matrigel or uncoated. Cells were cultured for 2–3 weeks in M4 at 37˚C to achieve again cell confluence onto the laminas. Cells were then observed under an inverted microscope (Carl Zeiss AG, Dublin, CA) and photographed under phase contrast to assess complete and uniform

**Table 1. Cornea´s donor data, including age, sex, days stored in Optisol until culture, and endothelial cell count.**

| Serial number | Age | Sex | Days to culture | Cell count/mm$^2$ | Cause of death |
|---|---|---|---|---|---|
| 1 | 22 | Male | 11 | 3063 | MVA |
| 2 | 16 | Female | 6 | 3496 | MVA |
| 3 | 25 | Male | 5 | 2991 | MVA |
| 4 | 31 | Female | 10 | 2705 | CAD |
| 5 | 35 | Male | 5 | 2608 | MVA |
| 6 | 38 | Male | 9 | 2759 | GSW |
| 7 | 30 | Male | 10 | 3279 | MVA |
| 8 | 33 | Female | 7 | 3010 | Liver failure |
| 9 | 24 | Female | 11 | 3151 | Acute cardiac crisis |
| 10 | 32 | Female | 8 | 2936 | Cardiogenic shock |
| 11 | 33 | Male | 9 | 2748 | Suicide |
| 12 | 24 | Male | 6 | 3012 | Cardiomyopathy |
| 13 | 24 | Female | 10 | 2541 | Peritonitis |
| 14 | 20 | Male | 10 | 2941 | MVA |
| 15 | 27 | Male | 10 | 2500 | Overdose |
| 16 | 20 | Female | 8 | 3940 | MVA |

MVA: Motor vehicle accident. CAD: Coronary artery disease. GSW: Gunshot wounds.

colonization of the decellularized lamina. The cell number in a 0.1-mm square was counted at four different sites after staining with trypan blue for 2 minutes or alizarin red for 15 minutes.

To demonstrate pumping function of human cultured CEC, we performed immunofluorescence against Na$^+$/K$^+$ ATPase in cultured cells. Briefly, cells were fixed in ethanol 100˚ for 5 min at 4C, and anti- Na$^+$/K$^+$ ATPase (Santa Cruz Biotechnology, Santa Cruz, CA, USA) at 1:400 was incubated 90 min at RT and then anti-mouse-biotin and avidin-FITC (Vector Laboratories, Burlingame, CA, USA) were incubated for 1 h each at RT. Nuclei were stained with DAPI, and cultures observed under an inverted epifluorescence confocal microscope (Leica). For positive controls, HeLa cells were used. For negative controls, cultures incubated the same way but omitting the primary antibody were used.

## Electron microscopy

For electron microscopy observation, eight endothelial cell seeded laminas were embedded in a fixative consisting of 2% paraformaldehyde and 2% glutaraldehyde (Wako Pure Chemicals) in 0.1 M cacodylate buffer at pH 7.4 for 1 h at 4˚C. Next, they were washed, treated with 1% osmium tetroxide in 0.1 M cacodylate buffer and embedded in epoxy resin for standard transmission electron microscopy. Ultrathin sections were stained with uranyl-lead and examined in a JEOL JEM 1010 transmission electron microscope (JEOL USA, Inc., MA, USA) to assess adhesion of endothelial cells to decellularized lamina and possible morphological or differentiation changes in the endothelial cells.

## Transplantation of decellularized corneal stroma lamellae with cultured passage 1 corneal endothelial cells in a rabbit model

To assess the feasibility of these *ex vivo* endothelial grafts in an *in vivo* model, 14 adult New Zealand White rabbits (Granja San Bernardo, Navarra, Spain) weighing 4 to 5 kg were used. The animals were treated in accordance with the ARVO Statement for the Use of Animals in Ophthalmic and Vision Research.

The animals were anesthetized with a combination of intramuscular ketamine (35 mg/kg) and xylazine (10 mg/kg) and also topically with a mixture of tetracaine 0.1% and oxybuprocaine hydrochloride 0.4% eye drop solution (Colicursi Anestésico Doble, Alcon Cusi, Barcelona, Spain). The rabbits were divided into two groups: a tissue engineered endothelial graft (TEEK) group, using a decellularized lamina disc FNC coated and with cultured hCECs; and a control group (controls) in which the decellularized lamina was FNC coated but devoid of endothelial cells. Each group comprised 7 eyes of 7 rabbits. Whether the graft was from the TEEK group or control group was unknown for the surgeon and for the individuals who assessed the outcomes.

The surgical approach was similar to DSAEK performed in a clinical setting. An extracapsular crystalline extraction was performed, which allowed a deeper anterior chamber and thus safe performance of an endothelial keratoplasty. Crystalline lens extraction was combined with endothelial grafting and was performed before descematorrhexis. An anterior chamber maintainer was positioned to stabilize the anterior chamber, and the DM in the central 9 mm was peeled off using a reverse Sinskey hook. As rabbits endothelium can regenerate from the periphery even after 9mm diameter central destruction [20,21], a follow up of only 4 weeks was performed to ensure no time for rabbit endothelial regeneration occurred.

Through the 5-mm corneal incision used for crystalline extraction, the graft was pulled through with a Busin glide and a prolene 10/0 suture (for details see [22]). An asymmetric "F" mark was drawn on the nonendothelial side of the graft to determine the correct orientation. The F mark was done over the stromal surface, opposite to the endothelialized surface using a dermatological ink pen by folding the graft when inserted into the injection glide. After the graft insertion and closure of the main incision with a 10/0 nylon suture, sterile 22μm filtered air was injected into the anterior chamber to attach the graft to the posterior surface of the cornea. After 10 minutes of supraphysiological pressure, the air was reduced to prevent pupillary block.

Postsurgical treatment consisted of a subtenon corticoid trigon and also topical eye drops of cyclopentolate (Cicloplejico, Alcon Cusi, Spain) twice a day for one week, subcutaneous tramadol 5 mg/kg twice a day for one week, and a topical combination of tobramycin 0.3% and dexamethasone 0.1% (Tobradex, Alcon Cusi, Spain), twice a day for the entire experimental period of one month. The rabbits were fed hay *ad libitum* in addition to chow every day during the study.

All the eyes were evaluated three times per week with slit lamp microscopy with a topical double anaesthetic colirium tetracaine 0.1% and oxybuprocaine 0.4% (Colircusi, Alcon Cusi, Spain) treated animals. Eyes were photographed for 4 weeks by an independent ophthalmologist and in a masked fashion. Corneal clarity and edema were scored on a scale of 0 to 4, as previously described in a rejection model [23,24]. The scoring system used was as follows: For clarity: 0, clear cornea; 1, slight haze; 2, increased haze but anterior chamber structures still clear; 3, advanced haze with difficult view of the anterior chamber; 4, opaque cornea without view of the anterior chamber. For edema: 0, no stromal or epithelial edema; 1, slight stromal thickness; 2, diffuse stromal edema; 3, diffuse stromal edema with microcystic edema of epithelium; 4, bullous keratopathy. Other ocular complications noticed were also recorded.

## Histological examination

The rabbits were euthanized 4 weeks after transplant with an overdose of sodium pentobarbital injected under deep anesthesia. The corneas were excised and either fixed in 4% formaldehyde and included in paraffin for histological examination or frozen directly in isopentane for molecular biology studies. Paraffin-embedded sections were cut at a 8 μm thickness from the

corneal apex and placed on silane-coated microscope slides, stained with hematoxylin-eosin, and observed with light microscopy (Leica BioSystems, Buffalo Grove, IL, USA) to measure corneal thickness with Image J image analysis software and to assess leukocyte infiltration into the recipient cornea.

To demonstrate human CEC colonization of the rabbit´s corneas, immunohistochemistries against human-specific ribonucleoproteins and human-specific mitochondria were performed. Briefly, autofluorescence was quenched with NaBH$_4$ for 30 min, and then an antigens´ retrieval step at pH 6 for 40 min at 98˚C was performed. Then anti-human nuclei antibody (Merck-Millipore, Billerica, MA, USA) at 1:100 and anti-human mitochondria (Merck-Millipore) at 1:50, and anti-mouse FITC (Vector Laboratories, Burlingame, CA, USA) were incubated with the slides. Nuclei were stained with DAPI. Confocal microscopy (Leica) was used to take photographs at high resolution and magnification. For positive controls, sections of human pancreas were used. Cells of the rest of the transplanted corneas were used as negative controls.

## Polymerase chain reaction analysis

To further ensure human cells had seeded the corneal lamina, polymerase chain reaction (PCR) was performed. Genomic DNA for the PCR analyses was obtained from the rabbits' whole corneas by phenol chloroform extraction after a proteinase K digestion method. The DNA concentration and purity were determined by optical density using a NanoDrop spectrophotometer (NanoDrop Technologies, Thermo Scientific, Wilmington, DE, USA) followed by standard DNA PCR analysis (ProofStart DNA Polymerase, Qiagen Sciences, Germantown, MD, USA). To confirm the presence of hCECs, we performed PCR using primers for two ubiquitous genes. Primers specific for the human β2-microglobulin gene were used to specifically amplify human DNA: huβ2Micro (F): `5' CAG GTT TAC TCA CGT CAT CCA GC 3'`; huβ2Micro (R): `5' TCA CAT GGT TCA CAC GGC AGG C 3'`. For positive controls of the PCR reaction, amplification of the highly conserved housekeeping gene β-actin was performed using the primers β-actin (F): `5'GTG ACG AGG CCC AGA GCA AGA G 3'` and β-actin (R): `5'ACG CAG CTC ATT GTA GAA GGT GTG G 3'`. PCR conditions were 95˚C x 15 min for one cycle followed by 95˚C x 1 min / 65˚C x 1 min / 72˚C x 1 min for 31 cycles [25,26]. For negative controls, no DNA reactions were used. For positive controls, human DNA from adipose tissue was used. The 235-base pair (bp) amplified fragments of human β2-microglobulin or 122 bp β-actin were separated on 2% agarose gel.

## Statistical analysis

Clinical clarity and edema index scores and average corneal thickness were compared with the Mann–Whitney U test between the TEEK and the control groups. The *P* value for statistical significance in this evaluation was set to 0.05.

## Results

### Development of human TEEK

Corneal laminas were successfully completely decellularized and their structure was maintained as demonstrated by staining with hematoxylin and eosin or DAPI (Fig 1A and 1B). Graft transparency was assessed by placing the lamellae on a back-lit chart as in the He group study [12], showing optimal transparency (Fig 1C).

Cultured human endothelial cells from 16 different donors (Table 1) were maintained in culture tissue plastic for 4 weeks maintaining their typical hexagonal morphology (Fig 1D).

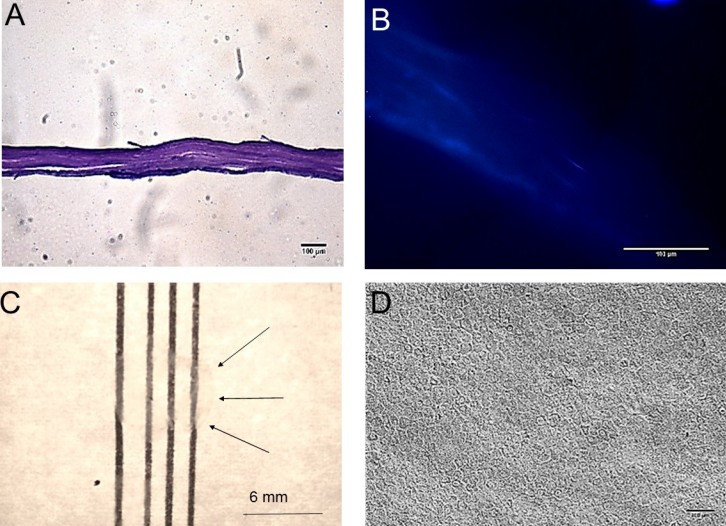

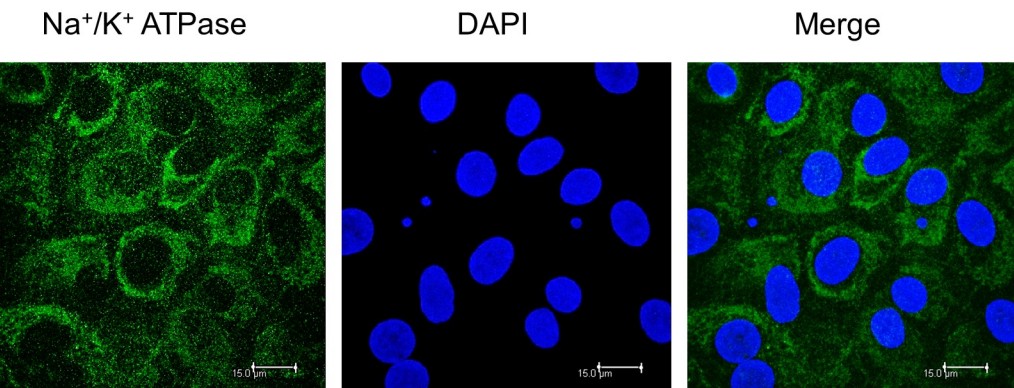

**Fig 1. hCEC culture characterization.** (A) Microscopic image of haematoxylin and eosin staining confirmed that the corneal stroma structure of decellularized corneal sheets was well maintained. (B) Lack of DAPI staining demonstrated the absence of cells in the scaffold. (C) Macroscopic image of decellularized lamina over a grid showing 90% transparency. (D) Phase contrast image of cultured hCECs isolated from donor corneas, showing the hexagonal morphology at passage 0. (E) Demonstration of cultured hCEC pumping function by Na⁺/K⁺-ATPase pump immunohistochemistry and confocal microscopy. Nuclei are stained with DAPI.

Cultured human endothelial cells conserved their characteristic pumping function in vitro, as demonstrated by active $Na^+/K^+$-ATPase pump (Fig 1E).

When they reached confluence, they were passaged over the decellularized lamina at $3 \times 10^3$ cells/mm$^2$ either uncoated or coated with Matrigel, denatured collagen IV (1% gelatin) or FNC (Fig 2A). The hCECs showed the ability to repopulate differently coated laminas and were able to form a confluent monolayer (Fig 2A). Cells seeded over gelatin-coated laminas demonstrated poor adherence, whereas cells seeded over a Matrigel coating showed a tendency to form a double layer. Cells repopulating laminas without any coating demonstrated good but poorer adherence than FNC-coated ones. In general, cells seeded by 10 min centrifugation showed poor adhesion. Electron microscope images of in vitro experiments confirmed all these results (Fig 2B). Collagen fiber bundles maintained their parallel pattern in the decellularized lamina, which also showed no remaining keratocytes (Fig 2B). FNC-coating laminas

A

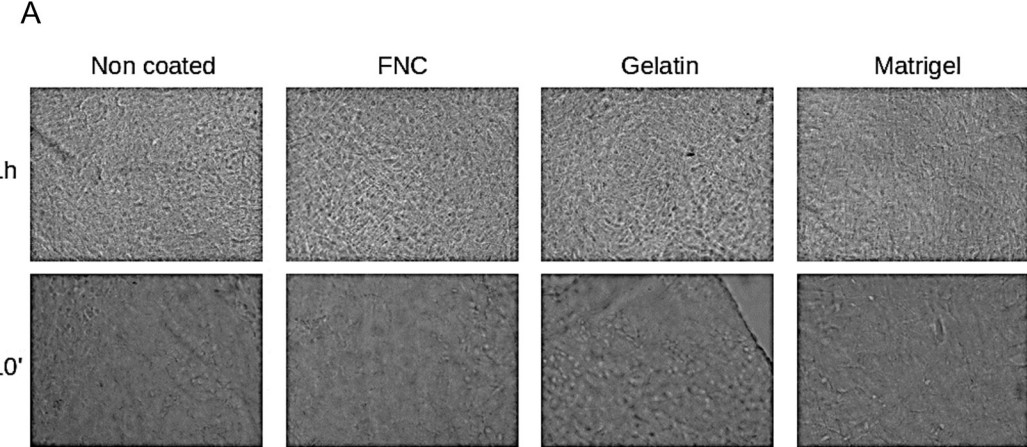

B

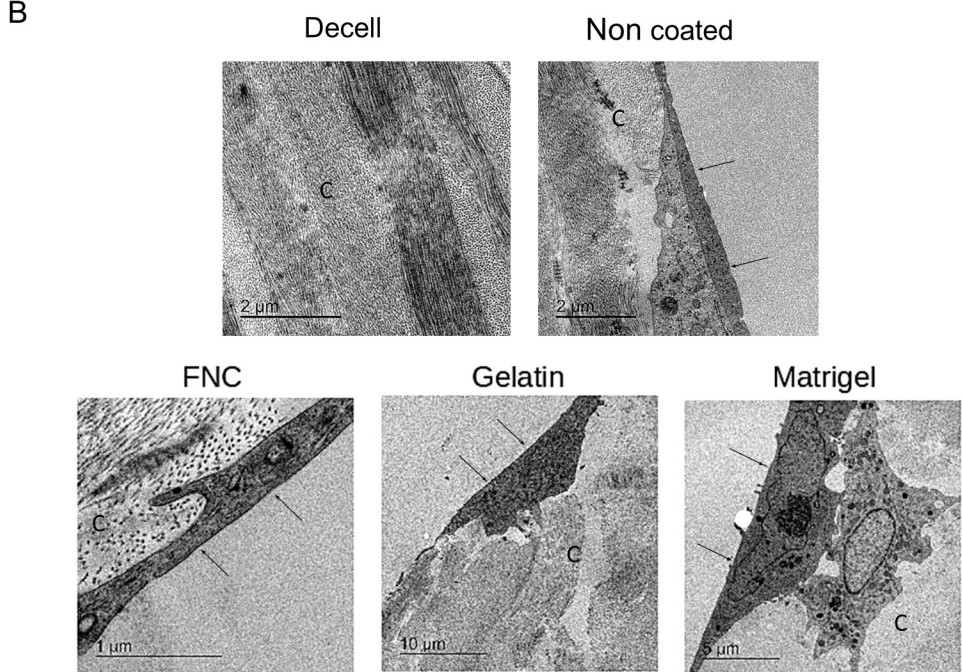

**Fig 2. hCEC colonized laminas characterization.** (A) Phase contrast images of corneal laminas with various coatings and repopulated with hCECs. Sheets were subjected to 10 minutes or 1 hour of centrifugation. (B) Electron microscopy images of various coated laminas after 1 hour of centrifugation. Note the perfectly aligned collagen bundle fibres in the decellularized lamina (Decell). Also, note the flat endothelial morphology and filopodia development between parallel collagen fibres in the FNC-coated sheet in comparison with double-layered cells in uncoated and Matrigel-coated sheets, and the poor adhesion in the gelatine-coated and uncoated sheets. C: Collagen fibres. Arrows: endothelial cells.

showed good cell-to-cell and cell-to-substrate interactions with hemidesmosomes and filopodia formation, respectively. Adhesion was better when the laminas were centrifuged for 1 hour. Noncoated laminas showed poor adhesion to decellularized stroma and developed fewer cell interactions. Gelatin-coating laminas did not attain good adhesion of hCECs. Interestingly,

the Matrigel coating promoted differentiation of hCECs into two cell types, one more similar to hCEC and other with fibroblastic characteristics (Fig 2B). The FNC coating maintained the hCEC natural flat morphology; their embedding into the lamina collagen fibers could be observed, and it showed the best adhesion and morphology (Fig 2B). These latter microscopic findings were similar to those in normal corneal endothelial cells *in vivo*. Subsequent experiments for *in vivo* applications were then performed with FNC coating and 1 h centrifugation. The hCEC on the grafts formed a monolayer and had a consistent size and a polygonal shape (n: 16). Mean cell density was 2300 cells/mm$^2$ confirmed by optical microscopy and Image J analysis prior to transplantation (Fig 2A). A mean of four grafts could be generated from 1 donor cornea hCECs in this way.

## Clinical observations after transplantation of decellularized lamina into rabbits

Slit lamp examination showed that all grafts from both groups attached to the posterior surface of the recipient rabbit cornea. Representative anterior segment photographs at day 3 and day 30 of the same animals are shown in S1A Fig. Iris suffered atrophy and is seen as a white band in all eyes. Corneal edema developed after surgery in both groups. The edema decreased from week 1 to week 4 in the TEEK group, whereas the edema was maintained for 4 weeks in the control group. The mean corneal edema index at 4 weeks was 2.6 in the control group and 1.3 in the TEEK group (n = 6; $P$ = .02) (S1B Fig). Similarly, transparency recovered gradually from week 1 to week 4 in the TEEK group, whereas it did not in the control group. The mean corneal haze index at 4 weeks was 3.7 in the control group and 1.8 in the TEEK group (n = 6; $P$ < .01) (S1C Fig). One rabbit from each group was excluded from analysis; the control group eye showed endothelialization of the lamina in histological examination and had a completely different recovery fashion than the rest of the control group. On the other hand, difficulties during insertion, and possibly upside-down tissue adhesion occurred in one eye of the TEEK group; that eye had persistent edema throughout the exams and was also excluded.

## Histological examination

Fig 3A and 3B show the results of the histological examination with light microscopy of the corneas at 28 days. Decellularized lamina grafts were colonized by stromal keratocytes at the peripheral side of the graft but not at the central part at 1 month after transplant (compare Fig 3A central versus peripheral). Stromal edema was observed histologically in both groups, but mean corneal thickness measured after histological preparation was significantly lower in the TEEK group than in the control group (747 vs. 1147 microns, respectively; $P$ = .02, Fig 3B). The posterior surface of the decellularized lamellae was covered with a cell monolayer in the TEEK group but not in the control group (Fig 3A), except for one case, which was excluded from analysis as previously noted. No leukocyte infiltration was observed in any cornea by H&E staining and histological exam.

To demonstrate that human CEC had colonized the rabbit´s cornea, anti human-specific ribonucleoprotein and antihuman-specific mitochondria immunofluorescence was performed in the corneas of transplanted rabbits at 4 weeks. Fig 4A shows in fact that all endothelial cells were of human origin.

## Polymerase chain reaction analysis

PCR analysis confirmed further that the endothelial cells were of human origin, given human β2-microglobulin was amplified in every rabbit cornea in the EK group (Fig 4B). As expected, no positive amplification of human cells was encountered in the control group (S2 Fig).

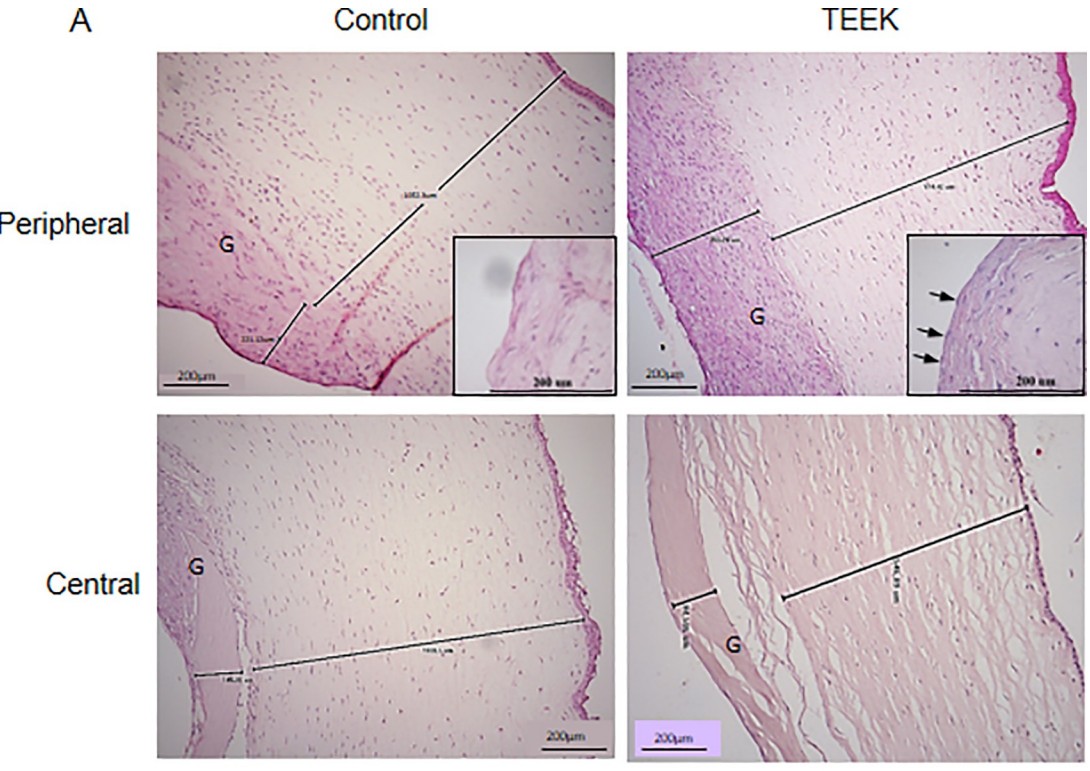

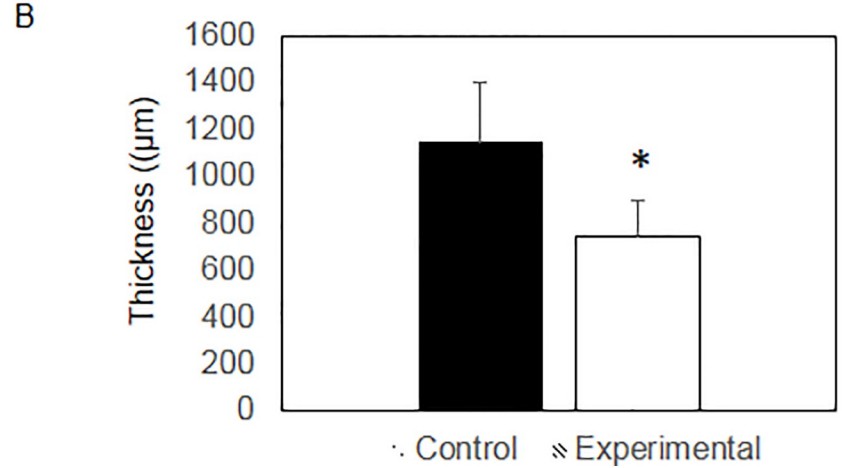

**Fig 3. Histological results.** Haematoxylin-eosin histological images of control and experimental (TEEK) corneas both at a peripheral section and at a central section of the cornea. Note the increased thickness of the control due to increased oedema. Graft thickness is also increased in both groups (G), and the experimental corneas shows endothelial cells. Insets show higher magnification demonstrating endothelial cells in the TEEK group (arrows). Also note the absence of leukocyte infiltration in every cornea. (B) Corneal thickness measurement graph showing decreased corneal thickness in the TEEK group at 4 weeks after the transplant. Data are given as mean±SD Asterisk indicates statistical significance at $P \leq .05$.

## Discussion

EK is the most common keratoplasty procedure performed in developed countries, and DSAEK is the predominant endothelial keratoplasty due to its simpler donor preparation and

A

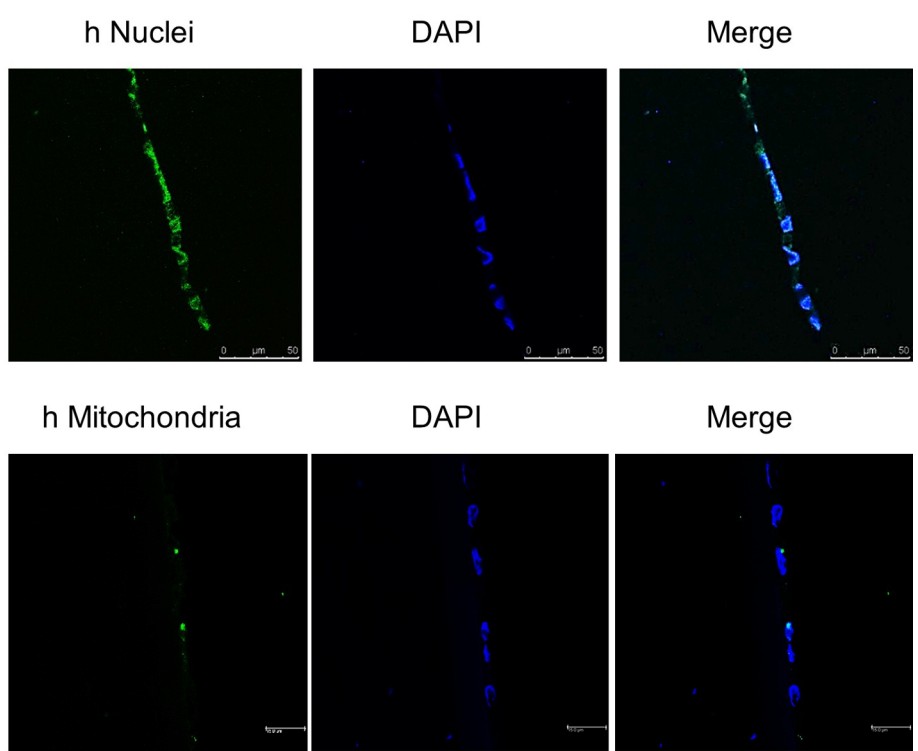

B

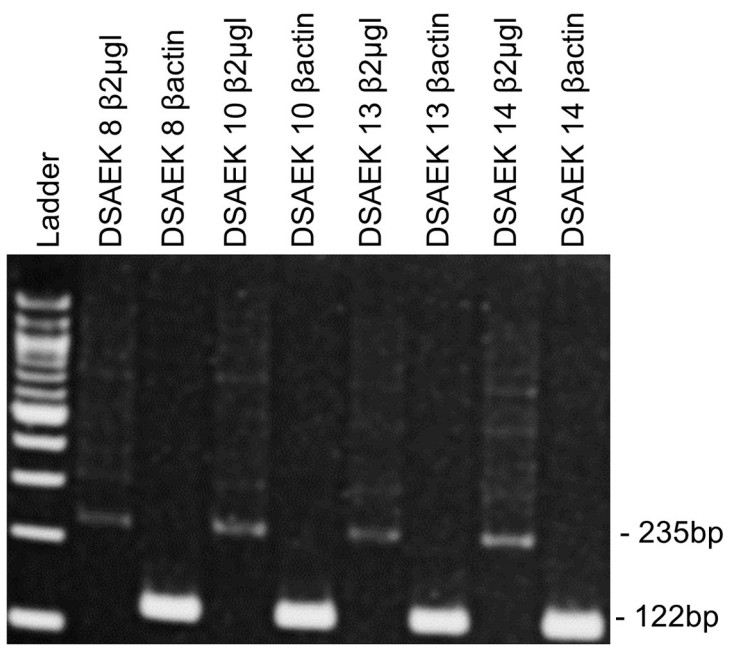

Experimental rabbits

**Fig 4. Human CEC corneal colonization.** A: Immunofluorescence for anti human-specific ribonucleoprotein and anti human-specific mitochondria in hCEC transplanted rabbit´s corneas at 4 weeks. Note merge of green (anti-human nuclei) and DAPI in every cell of the endothelium. B: PCR amplification of the housekeeping gene β-actin at 122 bp and the human specific gene β2-microglobulin in experimental rabbits' corneas. Every experimental cornea shows 235 bp β2-microglobulin amplification.

manipulation [27]. The present study showed that the use of decellularized stromal discs reconstructed with human corneal endothelial cells can be handled as TEEK donor grafts, and effectively reduced corneal edema and increasing transparency in an animal model of corneal endothelial injury.

Several carriers have been proposed to deliver corneal endothelial cells into an anterior chamber, including synthetic matrix [28–31] or xenogenic tissue [17,32,33]. However, human corneal stroma-derived carriers pose several advantages [12,15]. Specifically, the advantages of decellularized corneas as hCEC carriers are the absence of immunologic response as seen in the histological sections, and in stromal human transplantation [9], the maintenance of the ECM proteins intact and thus with suitable mechanical strength [34], the ability to provide a good scaffold to promote a good endothelial count with appropriate differentiation of hCEC, the maintenance of transparency, the ease of manipulation, the possibility of long term storage of the frozen tissue, and lack of ethical concerns as discarded corneas for transplant could be used. Moreover, these carriers can be readily provided by eye banks from discarded donor corneas with insufficient endothelial cell count.

The process to prepare human laminas from donor stroma using microtome cut [15,16] or femptosecond laser [12] has been previously assessed in the laboratory and has shown sufficient EC density, which retains expression of the functional markers Na$^+$/K$^+$- ATPase and ZO-1 while remaining transparent and thin and retaining the biomechanical properties similar to those of normal corneas [35]. Optimization of the best extracellular matrix coating of the lamina to promote hCEC adhesion and proliferation while maintaining proper cellular morphology and pumping function was critical to the success of our study. To the best of our knowledge, we have conducted the first study using several decellularized stromal laminas from the same donor for human corneal endothelial transplantation in an animal model.

In the animal model, the postoperative recovery was not complete at 1 month, given the corneal thickness had not returned to normal values but was statistically significant in comparison with the control group. There were clinically significant differences in corneal opacity and corneal thickness. A longer follow-up would probably have yielded further recovery of corneal clarity, but due to the intrinsic proliferative capacity of rabbit endothelium (in contrast to human endothelium), which could affect the results, the rabbits were euthanized at 30 days. In fact, one control eye was excluded from the study for that reason. The presence of endothelial cells covering the posterior corneal surface was shown in all the experimental cases at the end of the postoperative period, and these cells were of human origin. The slower and more incomplete restoration of corneal thickness has also been reported in a DSAEK model using non decellularized corneas [16]. Complete recovery after 4 weeks was also previously reported when leaving an intact Descemet Membrane, with the disadvantage of requiring one donor cornea for each recipient [35].

We found that as the host cornea recovered transparency, the donor lenticule showed a slight haziness. The same behavior of the lenticule after stromal enhancement for keratoconus with complete corneal transparency restoration 3 months postsurgery has been observed in patients after one month. The advantages of decellularized corneas as hCEC carriers include the absence of an immunologic response in the histological sections of this study; and in stromal human transplantation [9], the extracellular matrix proteins are maintained intact and

thus have a suitable mechanical strength [34]. In addition, good hCEC colonization and morphology are seen, transparency is maintained, delivery is similar to current techniques, there is the possibility of long-term storage of the frozen tissue and there is a lack of ethical concern given corneas discarded for transplant can be used.

The possibility of coating with hCEC the posterior surface of a decellularized lamina together with our demonstrated colonization and differentiation of extraocular stem cells into stromal keratocytes [9,12] and corneal epithelial cells (Casaroli-Marano et al. submitted, 2019, our own unpublished observations) brings as much closer to the development of a full corneal substitute using decellularized human stroma as a scaffold, which is a long-term goal assuming progressive success using other scaffolds [36,37].

Our study provides preclinical data showing that decellularized corneal stroma reconstituted with human corneal endothelial cells might be a surgical option for increasing availability of suitable tissue for endothelial disease, allowing more patients to be treated with the material obtained from one donor, both for endothelium and stroma. This carrier, which behaved as DSAEK grafts do, would simplify the adoption of the technique by most corneal surgeons who are used to endothelial keratoplasty. About four 150-micron-thick laminas can be obtained and seeded with endothelial cells from a single donor cornea. The carriers could be obtained from the cornea from which the endothelial cells are retrieved, or from a cornea discarded for transplant.

However, the use of human corneas poses some drawbacks regarding homogenization of the tissue. Future preclinical validation studies should be performed to assess whether the depth of the lamina cut (anterior, intermediate or posterior stroma) or the age of the donor cornea influences the mechanical or optical quality of the carrier or modifies the process of endothelial cell seeding.

Refinements in the method, using enhancers of the density and quality of hCECs, appears advisable to further improve the corneal edema resolution we observed. The use of Rock-inhibitors has been shown to improve the regenerating capacity of *in vivo* endothelial tissue [25]. A longer follow-up period with more cases would be necessary to draw stronger conclusions prior to clinical trials. Although decellularized corneal stroma poses no immunological threat, as has been demonstrated in corneal stroma transplantation [19], human endothelial cells seeded into the carriers for transplantation into an animal model behave as a xenograft, and therefore even better results would be expected if implanted in humans instead. Lastly, up to date, the coatings that favor endothelial cell adhesion such as FNC, do not have GMP/clinical grade, and could not be used in clinical practice, so alternative coating should be investigated.

## Conclusions

In conclusion, we have demonstrated that grafting of decellularized stroma carriers seeded with endothelial cells *ex vivo* can be a readily translatable method to improve visual quality in corneal endothelial diseases, but further refinements in the technique are needed.

## Supporting information

**S1 Fig. In vivo ophthalmic results.** (A) Representative macroscopic live images of rabbit corneas subjected to lens removal and corneal endothelial damage 1 day and 4 weeks after surgery. Note the decreased haze in the experimental rabbit cornea at 4 weeks. Magnification 3x. (B and C) Progression of corneal oedema (B) and haze (C) graphs after 1 and 4 weeks. Data is shown as mean±SD. Asterisks indicate statistical significance at $P \leq .05$.
(TIF)

**S2 Fig. Absence of human cells in non-transplanted rabbits.** PCR amplification of the housekeeping gene β-actin at 122 bp and the human specific gene β2-microglobulin in control rabbits' corneas. No control shows 235 bp β2-microglobulin amplification, whereas they do for the housekeeping β-actin gene.
(TIF)

## Acknowledgments

The authors wish to acknowledge the Experimental Surgery Department of La Paz University Hospital for anesthetic help. The authors also thank Juliette Siegfried and her team at servingmed.com for English editing.

## Author Contributions

**Conceptualization:** Francisco Arnalich-Montiel, Sherezade Fuentes-Julián, Gary Peh, Jodhbir S. Mehta, Maria P. De Miguel.

**Data curation:** Francisco Arnalich-Montiel, Sherezade Fuentes-Julián, Maria P. De Miguel.

**Formal analysis:** Francisco Arnalich-Montiel, Adrian Moratilla, Sherezade Fuentes-Julián, Veronica Aparicio, Marta Cadenas Martin, Laura Porrua, Maria P. De Miguel.

**Funding acquisition:** Francisco Arnalich-Montiel, Maria P. De Miguel.

**Investigation:** Francisco Arnalich-Montiel, Adrian Moratilla, Sherezade Fuentes-Julián, Veronica Aparicio, Marta Cadenas Martin, Laura Porrua, Ane Pérez-Sarriegui, Maria P. De Miguel.

**Methodology:** Francisco Arnalich-Montiel, Adrian Moratilla, Sherezade Fuentes-Julián, Veronica Aparicio, Marta Cadenas Martin, Gary Peh, Jodhbir S. Mehta, Khadijah Adnan, Laura Porrua, Ane Pérez-Sarriegui, Maria P. De Miguel.

**Project administration:** Maria P. De Miguel.

**Resources:** Maria P. De Miguel.

**Software:** Adrian Moratilla.

**Supervision:** Gary Peh, Jodhbir S. Mehta, Maria P. De Miguel.

**Validation:** Francisco Arnalich-Montiel, Adrian Moratilla, Veronica Aparicio, Gary Peh, Jodhbir S. Mehta, Khadijah Adnan, Laura Porrua, Ane Pérez-Sarriegui, Maria P. De Miguel.

**Visualization:** Adrian Moratilla.

**Writing – original draft:** Francisco Arnalich-Montiel, Sherezade Fuentes-Julián, Maria P. De Miguel.

**Writing – review & editing:** Francisco Arnalich-Montiel, Adrian Moratilla, Sherezade Fuentes-Julián, Veronica Aparicio, Marta Cadenas Martin, Gary Peh, Jodhbir S. Mehta, Khadijah Adnan, Laura Porrua, Ane Pérez-Sarriegui, Maria P. De Miguel.

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
