## [Decision Letter · Decision Letter 0]

21 Aug 2019

PONE-D-19-14728

Treatment of corneal endothelial damage in a rabbit model with a bioengineered graft using human decellularized corneal lamina and cultured human corneal endothelium

PLOS ONE

Dear Dr De Miguel,

Thank you for submitting your manuscript to PLOS ONE. After careful consideration, we feel that it has merit but does not fully meet PLOS ONE’s publication criteria as it currently stands. Therefore, we invite you to submit a revised version of the manuscript that addresses the points raised during the review process.

I believe the reviewers comments to be pertinent. Proof that the transplanted human cells have remained in place and at a similar number following transplantation for the 4 weeks should be provided. I appreciate that PCR against human DNA was included but as the reviewer indicates this does not give a readout on how many cells were retained.

You might argue that the control did not show the same effect so it must be the presence of the human cells. However, the presence of the human cells might stimulate the endogenous cells to migrate/proliferate more aggressively.

Perhaps the inclusion of in vivo endothelial cell morphology at multiple time points could help (i.e. at least every 7 days or more). This way one might  follow changes in cell morphology or not, the latter suggesting little change. Otherwise immunohistology against human antibodies or preferably Fluorescence in situ hybridization (FISH) using day 30 postmortem corneas.

Otherwise I found the article to well written and scientifically robust.

If you can address the above concern I support a resubmission of this study

We would appreciate receiving your revised manuscript by Oct 05 2019 11:59PM. To enhance the reproducibility of your results, we recommend that if applicable you deposit your laboratory protocols in protocols.io, where a protocol can be assigned its own identifier (DOI) such that it can be cited independently in the future. For instructions see: http://journals.plos.org/plosone/s/submission-guidelines#loc-laboratory-protocols

We look forward to receiving your revised manuscript.

Kind regards,

Che J. Connon

Academic Editor

PLOS ONE

Journal Requirements:

1. Please move Fig. 2 to a Supporting Information file

2. In your Methods section, please provide additional information regarding the donated tissues or organs used in the study. Please specify where the tissues come from. Please also specify whether the study involved the use of donated tissue/organs from any vulnerable populations. Examples of vulnerable populations include prisoners, subjects with reduced mental capacity due to illness or age, and children. If such a population was used, please ensure you have describe the population and justify the decision to use tissue/organ donations from this group. If not, please state in your Ethics Statement, 'None of the transplant donors were from a vulnerable population and all donors or next of kin provided written informed consent that was freely given.

Reviewers' comments:

Reviewer's Responses to Questions

**Comments to the Author**

1. Is the manuscript technically sound, and do the data support the conclusions?

Reviewer #1: No

2. Has the statistical analysis been performed appropriately and rigorously? 

Reviewer #1: Yes

3. Have the authors made all data underlying the findings in their manuscript fully available?

Reviewer #1: Yes

4. Is the manuscript presented in an intelligible fashion and written in standard English?

Reviewer #1: Yes

5. Review Comments to the Author

Reviewer #1: Authors used human decellularized corneal stroma as a career of bioengineered corneal graft and performed experiments in rabbit eyes. They concluded the experimental graft was effective in improving visual quality. However, the methods used here did not prove and support their conclusions.

Experimentally proliferated hCECs might not have enough pumping function. They need to prove it either in vitro or in vivo.

Rabbit endothelial cells regenerate very fast. 0.5mm gap between graft and host endothelial edge could be filled out quickly, probably within the observational period in this study. Authors need to show the improvement of the corneal edema/clarity was due to human corneal cells seeded, not from regenerated rabbit healthy cells. They only performed HE stain of the extracted grafts and did not show the endothelial cells covering the graft were of human origin. Using simple PCR indirectly proves the human cell presence but did not prove the full coverage and survival of hCECs seeded on the graft.

Authors measured corneal thickness in HE sections but it must be directly measured in vivo by pachymeter or AS-OCT.

6. PLOS authors have the option to publish the peer review history of their article (what does this mean?). If published, this will include your full peer review and any attached files.

Reviewer #1: No

---

## [Author Response · Author response to Decision Letter 0]

29 Oct 2019

PONE-D-19-14728

Treatment of corneal endothelial damage in a rabbit model with a bioengineered graft using human decellularized corneal lamina and cultured human corneal endothelium

PLOS ONE

Answers to editor and Reviewer´s comments:

Editors´ Comments

I believe the reviewers comments to be pertinent. Proof that the transplanted human cells have remained in place and at a similar number following transplantation for the 4 weeks should be provided. I appreciate that PCR against human DNA was included but as the reviewer indicates this does not give a readout on how many cells were retained.

You might argue that the control did not show the same effect so it must be the presence of the human cells. However, the presence of the human cells might stimulate the endogenous cells to migrate/proliferate more aggressively.

Perhaps the inclusion of in vivo endothelial cell morphology at multiple time points could help (i.e. at least every 7 days or more). This way one might follow changes in cell morphology or not, the latter suggesting little change. Otherwise immunohistology against human antibodies or preferably Fluorescence in situ hybridization (FISH) using day 30 postmortem corneas.

We have performed anti-human nuclei and anti-human mitochondria immunohistochemistries in sections of 30 day postmortem experimental rabbits corneas to show that in fact all endothelial cells are of human origin. This is now shown now in Figure 4A.

Otherwise I found the article to well written and scientifically robust. If you can address the above concern I support a resubmission of this study

Thank you very much for your kind words, hopefully now we have addressed the downfalls of our article properly.

1. Please move Fig. 2 to a Supporting Information file

Done, it is now labelled as SupplFigure1.

2. In your Methods section, please provide additional information regarding the donated tissues or organs used in the study. Please specify where the tissues come from. Please also specify whether the study involved the use of donated tissue/organs from any vulnerable populations. Examples of vulnerable populations include prisoners, subjects with reduced mental capacity due to illness or age, and children. If such a population was used, please ensure you have describe the population and justify the decision to use tissue/organ donations from this group. If not, please state in your Ethics Statement, 'None of the transplant donors were from a vulnerable population and all donors or next of kin provided written informed consent that was freely given.

None of the transplant donors were from a vulnerable population and all donors or next of kin provided written informed consent that was freely given. This has been revised and edited in the Materials and Methods manuscript´s section. In addition, a table with all information of the donated tissues has been included (Table 1).

This has been now shown in Suppl.Figure2

Reviewers' comments:

5. Reviewer #1: Comments to the Author

Authors used human decellularized corneal stroma as a career of bioengineered corneal graft and performed experiments in rabbit eyes. They concluded the experimental graft was effective in improving visual quality. However, the methods used here did not prove and support their conclusions.

Experimentally proliferated hCECs might not have enough pumping function. They need to prove it either in vitro or in vivo.

We have performed immunohistochemistry demonstrating the pumping function of proliferated hCECs, shown as active Na+/K+-ATPase pump in Figure 1E. We want to thank the reviewer for this comment, as the new data has strengthen considerably our manuscript.

Rabbit endothelial cells regenerate very fast. 0.5mm gap between graft and host endothelial edge could be filled out quickly, probably within the observational period in this study. Authors need to show the improvement of the corneal edema/clarity was due to human corneal cells seeded, not from regenerated rabbit healthy cells. They only performed HE stain of the extracted grafts and did not show the endothelial cells covering the graft were of human origin. Using simple PCR indirectly proves the human cell presence but did not prove the full coverage and survival of hCECs seeded on the graft.

We agree that the 0.5mm gap could have been filled between the observational period. However, we feel that we have not described our approach very well, as the whole center of the cornea was decellularized by scraping out the endothelial layer, so even if the gap was filled, the majority of the cornea still was devoid of endothelial cells, as demonstrated in the control rabbits. Also that was the reason for a follow up period of only 4 weeks, to ensure not enough time for rabbit endothelial regeneration of the central cornea.

Still, we agree that proof of human cell survival and coverage on the graft should be provided, so we have now performed anti-human nuclei and anti-human mitochondria immunohistochemistries in sections of 30 day postmortem experimental rabbits corneas to show that in fact all endothelial cells are of human origin. This is now shown now in Figure 4A. Again we want to thank the reviewer for this most helpful suggestion.

Authors measured corneal thickness in HE sections but it must be directly measured in vivo by pachymeter or AS-OCT.

We agree that a pachymeter should have been the method of election to measure the corneal thickness, but we could not purchase it from the beginning of the experiment. In fact, we have measures of several but not all rabbits by such method, and have compared such measurements with subsequent ones in HE sections. In all cases, post-fixation measurements were consistently about 85% of the in vivo ones, so we are positive HE measurements are in fact reliable and representative of the in vivo measures when the control and experimental groups are compared, as all of them were processed in the exact same way.

---

## [Editor Report · Decision Letter 1]

6 Nov 2019

Treatment of corneal endothelial damage in a rabbit model with a bioengineered graft using human decellularized corneal lamina and cultured human corneal endothelium

PONE-D-19-14728R1

Dear Dr. De Miguel,

We are pleased to inform you that your manuscript has been judged scientifically suitable for publication and will be formally accepted for publication once it complies with all outstanding technical requirements.

With kind regards,

Che J. Connon

Academic Editor

PLOS ONE

Additional Editor Comments (optional):

The main cause of contention has been successfully resolved by inclusion of data in Fig4.

Reviewers' comments:

None

---

## [Editor Report · Acceptance letter]

12 Nov 2019

PONE-D-19-14728R1 

Treatment of corneal endothelial damage in a rabbit model with a bioengineered graft using human decellularized corneal lamina and cultured human corneal endothelium 

Dear Dr. De Miguel:

I am pleased to inform you that your manuscript has been deemed suitable for publication in PLOS ONE. Congratulations! Your manuscript is now with our production department. 

With kind regards,

on behalf of

Dr. Che J. Connon 

Academic Editor

PLOS ONE